# Global irrigation contribution to wheat and maize yield

Xuhui Wang [1✉], Christoph Müller [2], Joshua Elliot[3,4], Nathaniel D. Mueller [5,6], Philippe Ciais [1,7], Jonas Jägermeyr [3,4], James Gerber [8], Patrice Dumas [9], Chenzhi Wang [1], Hui Yang[1,7], Laurent Li [10], Delphine Deryng [11], Christian Folberth [12], Wenfeng Liu [13], David Makowski [14], Stefan Olin [15], Thomas A. M. Pugh [15], Ashwan Reddy[16], Erwin Schmid [17], Sujong Jeong [18], Feng Zhou [1] & Shilong Piao [1,19,20]

Irrigation is the largest sector of human water use and an important option for increasing crop production and reducing drought impacts. However, the potential for irrigation to contribute to global crop yields remains uncertain. Here, we quantify this contribution for wheat and maize at global scale by developing a Bayesian framework integrating empirical estimates and gridded global crop models on new maps of the relative difference between attainable rainfed and irrigated yield ($\Delta Y$). At global scale, $\Delta Y$ is 34 ± 9% for wheat and 22 ± 13% for maize, with large spatial differences driven more by patterns of precipitation than that of evaporative demand. Comparing irrigation demands with renewable water supply, we find 30–47% of contemporary rainfed agriculture of wheat and maize cannot achieve yield gap closure utilizing current river discharge, unless more water diversion projects are set in place, putting into question the potential of irrigation to mitigate climate change impacts.

[1] Sino-French Institute of Earth System Sciences, Peking University, 100871 Beijing, China. [2] Potsdam Institute for Climate Impact Research, 14473 Potsdam, Germany. [3] University of Chicago and ANL Computation Institute, Chicago, IL 60637, USA. [4] Columbia University Center for Climate Systems Research, New York, NY 10025, USA. [5] Department of Ecosystem Science and Sustainability, Colorado State University, Fort Collins, CO, USA. [6] Department of Soil and Crop Sciences, Colorado State University, Fort Collins, CO, USA. [7] Laboratoire des Sciences du Climat et de l'Environnement, CEA CNRS UVSQ Orme des Merisiers, 91191 Gif-sur-Yvette, France. [8] Institute on the Environment, University of Minnesota, St. Paul, MN 55108, USA. [9] Centre International de Recherche sur l'Environnement et le Développement, Nogent sur Marne 94130, France. [10] Laboratoire de Météorologie Dynamique, Université Pierre et Marie Curie, 75005 Paris, France. [11] Climate Analytics, 10969 Berlin, Germany. [12] Department of Geography, Ludwig Maximilian University, 80333 Munich, Germany. [13] College of Water Resources and Civil Engineering, China Agricultural University, 100083 Beijing, China. [14] INRA, AgroParisTech, Université Paris-Saclay, UMR 211 Agronomie, Thiverval-Grignon 78850, France. [15] Department of Physical Geography and Ecosystem Science, Lund University, 22362 Lund, Sweden. [16] Department of Geographical Sciences, University of Maryland, College Park, MD 20742, USA. [17] Institute for Sustainable Economic Development, University of Natural Resources and Life Sciences, 1180 Vienna, Austria. [18] Department of Environmental Planning, Graduate School of Environmental Studies, Seoul National University, Seoul, Korea. [19] Key Laboratory of Alpine Ecology and Biodiversity, Institute of Tibetan Plateau Research, Chinese Academy of Sciences, Beijing 100085, China. [20] Center for Excellence in Tibetan Earth Science, Chinese Academy of Sciences, Beijing 100085, China. ✉email: xuhui.wang@pku.edu.cn

Over the next decades, the projected increase in global population and increasing demand for animal and food products will require substantial increases in global crop production[1,2]. Since the expansion of cropland areas upon forested lands[3,4] has a cascade of negative ecological consequences[5,6], sustainable intensification pathways of crop production systems are needed in order to minimize environmental impacts. The challenge of increasing crop yields is further complexed by climate change, which significantly affected the crop yield at regional to global scale[7–9]. Improving irrigation is a possible option to achieve higher yield levels in water-limited regions while improving the resilience of cropping systems to climate variability[10–14].

Despite the known importance of irrigation for cereal yields[10,15,16], the contribution of irrigation to yield increment at regional to global scales remains uncertain. Different assumptions taken by different researchers based on hydrological models[17,18] can result in estimates that substantially differ by a factor of two for the yield gains brought by irrigation (+40% in Rosegrant et al.[17] against +20% in Siebert and Döll[18]). With growing understanding that the benefit of irrigation on yield varies largely with climatic conditions[19], there is an urgent need to understand how contributions of irrigation to yield varies with climate at global scale. To address this research problem[10,11,16,20], two approaches have been developed, based on climate analogues (CAs) and on process-based crop models.

The CA approach is based on the analysis of census and survey-derived yield data, combined with classification of climatic zones and irrigation extent. Attainable yields are defined as the 95th percentile yields within a climate zone, and these are calculated including and excluding irrigated areas to define rainfed and irrigated attainable yields[10] (see Methods section). With this approach, the contribution of irrigation to yield under current technology can be estimated because it implicitly accounts for factors (e.g. climate conditions, soil properties, and crop varieties) interacting with irrigation. The spatial extrapolation of the derived attainable yields relies however on relatively simple climate indices (e.g. growing degree days and precipitation) and neglects explicit consideration of soil characteristics. These indices have limited ability to account for weather variations that have disproportionate effects on attainable yields, such as dry periods[21], heat-waves[22,23], and cold spells in the reproductive growth stage[24]. As an alternative to using census and farm-scale data, global gridded crop models provide simulations of yields with and without irrigation, over the entire globe. Models include mechanistic representations of crop growth at daily or sub-daily temporal resolution, benchmarked against field data and yield censuses[25–27], and most models include representations of the ways soil characteristics influence yield (e.g. water holding capacity). Generally speaking, the ensemble-mean of these models was shown to have good capability for assessing the impacts of climate variations on yield[8,28]. Yet, models do have limitations because they do not account for the diversity of crop varieties, management practices, irrigation technology, and soil properties[29], and thus have potential biases in estimating the magnitude of irrigation contribution to the yield in a specific location.

As the advantages of CA and gridded crop model approaches are complementary, we integrate the CA and the global crop model results in a common statistical framework to assess relative differences between irrigated and rainfed yield ($\Delta Y$, see Methods section). The CA attainable yield dataset comes from updated Mueller et al.[10] (see Methods section) and crop model results from the Global Gridded Crop Model (GGCM) inter-comparison project[30] (see Methods section), with 10 global gridded crop models. A proper integration of different approaches could lead to better estimates outperform either approach[31]. However, traditional regression-based method relies on strong assumptions of known driving factors[31]. Here, with Bayesian Model Averaging (BMA) method[32] (see Methods section), we reanalyze $\Delta Y$ for wheat and maize, in which we do not assume any driver to its spatial variations in prior. The performance of the reanalyzed $\Delta Y$ outweighs both the CA and GGCM approaches when evaluated against US statistical survey data. Our further spatial analyses reveal the spatial variability of $\Delta Y$ was driven more by regional patterns of precipitations than by that of evaporative demand. Finally, we explore the balance between irrigation demands to close the yield gaps and available river discharge for irrigation, showing that a large fraction of contemporary rainfed agriculture could not rely on irrigation to achieve yield gap closure and mitigate climate change impacts without hydrological engineering efforts like cross-basin water transfer.

## Results and discussion

**Reanalyzed $\Delta Y$.** First, we contested $\Delta Y$ from CA, GGCMs, and BMA reanalysis against an independent dataset of irrigated and rainfed yield over US based on county yield surveys from the US Department of Agriculture (referred to as gridded-US hereafter; Schauberger et al.[12]). As Fig. 1 shows, $\Delta Y$ from CA (squared bias of 0.2% for both wheat and maize) is less biased than that from GGCMs (squared bias of 4.1% for wheat and 3.8% for maize, respectively), being distributed on both side of the 1:1 line with gridded-US (Fig. 1a, d). Although $\Delta Y$ from GGCMs shows a systematic overestimation with respect to the gridded-US data (Fig. 1b, e), its spatial pattern is better correlated with observations than the CA results ($r = 0.67$ for wheat, $r = 0.72$ for maize in Fig. 1b, e compared to $r = 0.49$ for wheat, $r = 0.59$ for maize in Fig. 1a, d). The Bayesian estimate combining CA and GGCMs provides a more precise estimate of $\Delta Y$ than either of the methods taken separately, with good spatial variations ($r = 0.69$ for wheat, $r = 0.76$ for maize) and a lower bias (squared bias of 1.7% for wheat and 1.2% for maize) than GGCMs (Fig. 1c, f). Although BMA results largely reduce the overestimate of $\Delta Y$ by GGCMs, in a global latitudinal transect, BMA results show generally higher $\Delta Y$ estimates compared to the CA approach, especially for maize across all latitudes and for wheat at high and low latitudes (Fig. 2a, b and Supplementary Fig. 1).

At global scale, reanalyzed $\Delta Y$ is 34 ± 9% for wheat (mean ± s.d.) over contemporary harvested area[33] and 22 ± 13% for maize. This suggests that irrigation benefit wheat yields to a larger degree than maize yield. The contribution of irrigation to crop yields has however large spatial differences, shown in Fig. 2. For wheat (Fig. 2a), $\Delta Y$ varies by one order of magnitude across latitudes, being larger in semi-arid and subtropical regions (between 15 and 23°N, $\Delta Y > 50\%$) compared to temperate regions ($\Delta Y < 10\%$). In major wheat producing regions with temperate climate, such as the US, eastern Europe (Ukraine and western Russia) and the lower reach of the Yangtze river basin, yield increases from irrigation are limited, reflecting sufficient precipitation during the growing seasons (Supplementary Fig. 2). By contrast, in drier areas like the US Great Plains, the Mediterranean, Central Asia, northern China and Australia, $\Delta Y$ exceeds 50%. Interestingly, there are wet and warm regions (with annual precipitation larger than 1000 mm) showing a large positive $\Delta Y$, such as southwestern China and India. This can be related either to a phase difference between wheat growing season and the timing of monsoon rains, or a larger evaporative demand induced by higher temperatures. The latitudinal differences in $\Delta Y$ of maize are not as large as those of wheat (Fig. 2b). Large $\Delta Y$ of maize is found in semi-arid and summer dry regions around ~30°N, such as the US Great Plains, southern Europe and northwestern China, with also Brazil (mainly the Cerrado area) and South Africa. Although the potential for

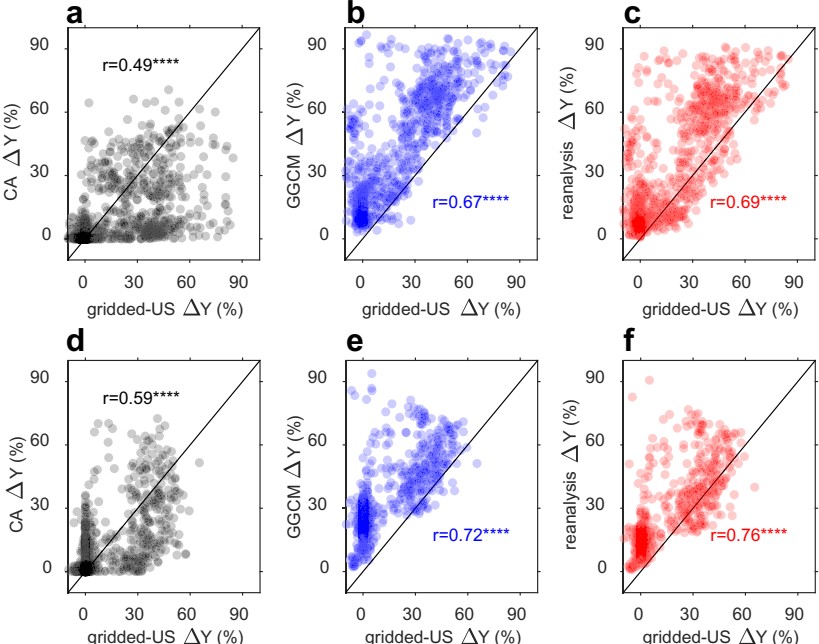

**Fig. 1 Comparison of irrigation contribution to yield ($\Delta Y$) estimated from statistics over conterminous United States (gridded-US) and from different approaches, for wheat (top panels) and for maize (bottom panels). a, d** $\Delta Y$ estimated from the climate analogue (CA) approach; **b, e** $\Delta Y$ estimated from global gridded crop models (GGCM); **c, f** $\Delta Y$ estimated from Bayesian model average (BMA). Details of different datasets and approaches can be found in the Methods section. $r$ indicates Pearson correlation coefficient between $\Delta Y$ estimated from gridded-US and other $\Delta Y$ estimates. **** indicates significant ($p < 0.01$) Pearson correlation (two-tailed tests, no adjustments).

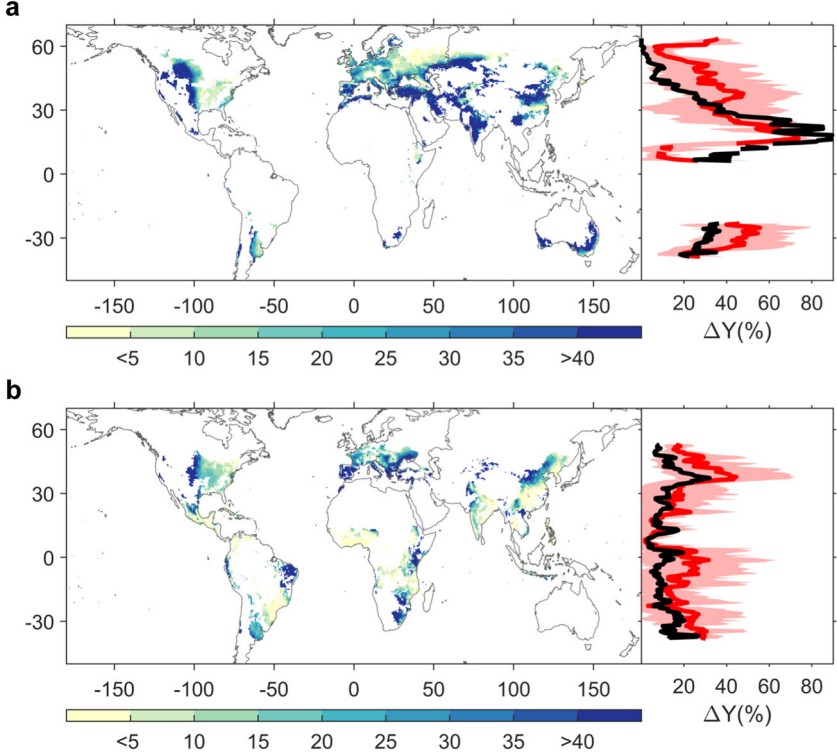

**Fig. 2 Spatial and latitudinal changes in $\Delta Y$ over contemporary growing area for wheat and maize. a** Wheat, **b** maize. The left panel represents spatial distribution of reanalyzed $\Delta Y$. The right panel shows latitudinal distribution of $\Delta Y$ for each one degree latitudinal band. The black curve shows $\Delta Y$ estimated from the climate analog (CA) and the red curve shows $\Delta Y$ estimated from the reanalysis, with shaded area indicates the range of uncertainty (1$\sigma$ standard deviation across models).

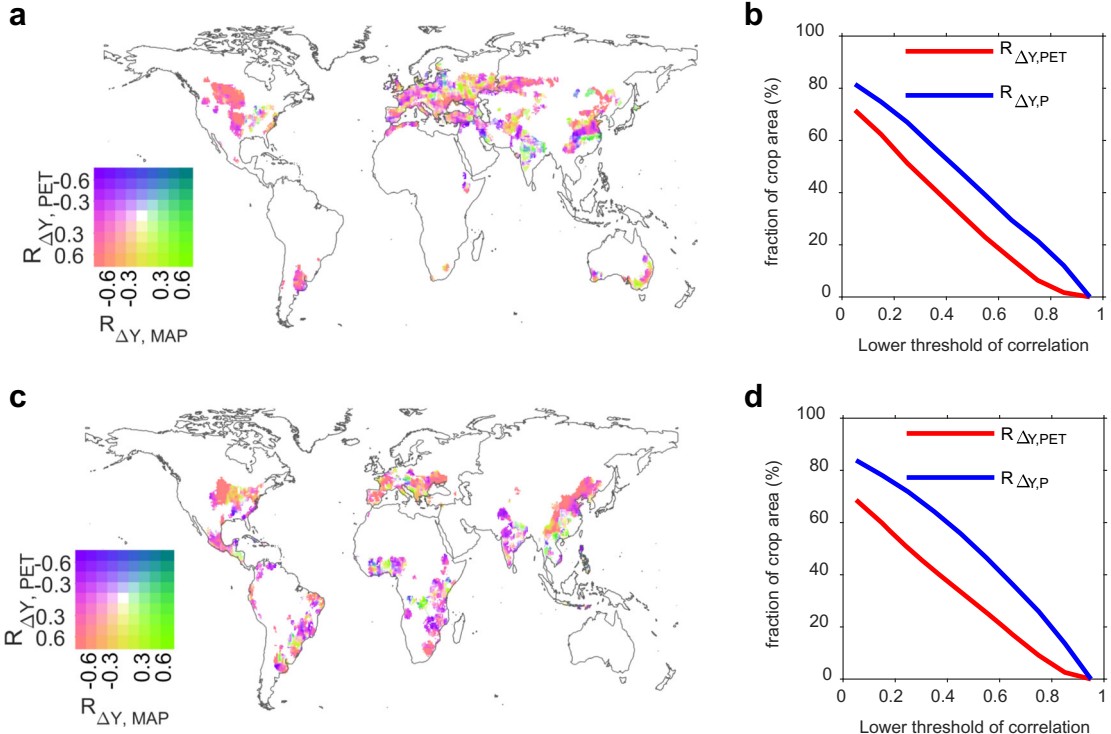

**Fig. 3 Partial correlation in the spatial domain between reanalyzed ΔY and climatic variables (potential evapotranspiration (PET) and mean annual precipitation (MAP)) for wheat (top panels) and for maize (bottom panels). a, c** Bivariate mapping for spatial distribution of the partial correlation coefficient between ΔY and PET ($R_{\Delta Y,PET}$) and that between ΔY and MAP ($R_{\Delta Y,MAP}$). **b, d** Percentage of cropland area where ΔY is controlled by PET or precipitation depending on the chosen threshold (x-axis) for the partial correlation coefficients.

yield increment over sub-Saharan Africa is generally high[10], the ΔY of maize over this region is low, indicating that yields are less constrained by water than by other factors, such as nutrients[10].

**Climatic drivers for spatial variations in ΔY.** Given that climate drivers for the spatial variations in ΔY could vary among different crops and regions, we performed partial correlation analyses between ΔY and climate variables for 3.5° by 3.5° moving windows (Fig. 3). We find that spatial variations of ΔY significantly correlated with mean annual precipitation over half of the crop area (62% for wheat and 66% for maize), while less crop area (49% for wheat and 48% for maize) is significantly correlated ($P < 0.05$) with mean annual potential evapotranspiration (PET, see Methods section). For both wheat and maize, the dominance or co-dominance of PET in ΔY are only found north of 40°N, such as Canada, the Northeast US, and Northeast China, while precipitation is dominant in spatial variations over all other regions. This implies that spatial variations in ΔY are more determined by spatial variations in climatic water supply, proxied by precipitation, rather than climatic water demand, proxied by PET. Using other algorithms to obtain PET or using mean annual temperature as the other proxy of water demand show similar results (Supplementary Figs. 3 and 4; see Methods section).

**The balance between the irrigation demand and supply.** Whether intensifying irrigation can realize the ΔY values depends also on available water resources. We thus calculated the irrigation requirement for reaching ΔY (see Methods section) and compared it with river discharge[34], which provides a limit to renewable freshwater supply for irrigating contemporary rainfed wheat and maize croplands. Specifically, two parameters are considered for harnessing runoff to increase irrigation, (1) the ΔY

low threshold which determines the minimum yield increment due to irrigation, above which irrigation is applied and (2) the maximum fraction of river discharge that can be used sustainably for irrigation without compromising the riverine ecosystems[11]. When considering a reasonable range for these two parameters (Fig. 4a; see Methods section), we found that 80–126 million ha of contemporary rainfed wheat and maize cropland do not have access to sufficient discharge to meet the irrigation demand (Fig. 4b and Supplementary Fig. 5). This area where more irrigation would be beneficial, but may not be achievable, represents 30–47% of the contemporary rainfed croplands of wheat and maize, considering different thresholds in water extraction. The largest areas with insufficient irrigation water supply from discharge alone are concentrated around 30°S and 30°N, including western US and Canada, circ-Black Sea, Central Asia, North and Northeast China, Argentina, South Africa, and southeastern Australia (Fig. 4b and Supplementary Fig. 6) with the largest deficit found in Australia exceeding 100 mm y$^{-1}$. Most of the African countries, where prevalence of undernourishment was highest today (Supplementary Fig 6), seem to have sufficient water supply to fulfill the irrigation needs (Fig. 4b), but may face substantial constraints from the governance level[35], which is important for long-term investments in irrigation infrastructure[36]. When comparing the irrigation demand with current river discharge[37] for major river basins where wheat and maize are grown (Supplementary Fig. 7), we also found large spatial heterogeneity in the balance between water supply and irrigation demand (Supplementary Table 1). The projected additional irrigation water requirements to fill the irrigation yield gap for wheat and maize represent <0.1% of river discharge in the Congo basin but would exceed the current river discharge of the Murray basin by a factor of three (Supplementary Table 1). Irrigation requirements exceed 20% of today's river discharge for one fifth of the

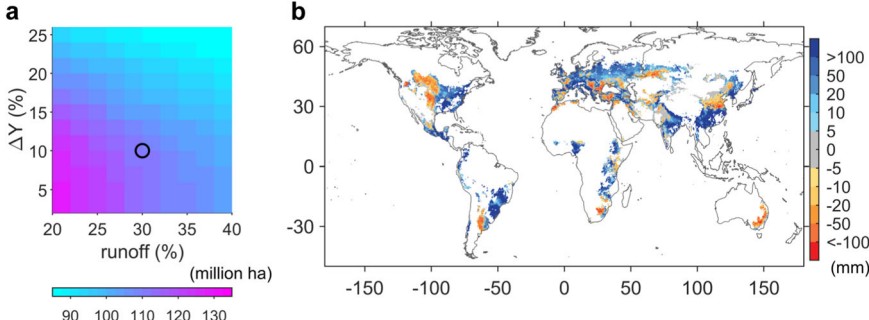

**Fig. 4 Relationship between irrigation demand estimated from the reanalysis for contemporary rainfed croplands of wheat and maize and available runoff resources. a** The amount of rainfed crop area when irrigation demand cannot be met with available runoff resources, according to different minimum threshold of $\Delta Y$ (y-axis) and maximum threshold of runoff consumption (x-axis). **b** The spatial distribution of the difference between irrigation demand and available runoff resources. The spatial pattern is determined with the minimum threshold of $\Delta Y$ for demanding irrigation is 10% and the maximum usage of runoff is 30% (corresponding to the black circle in **a**). See Supplementary Fig. 5 for spatial pattern of different thresholds of maximum runoff usage.

basins (Don, Huai, Tigris and Euphrates, Yellow River, Ural), highlighting the grand challenge of fully realizing the potential of irrigation to increase crop yield globally. If further considering the fact that today's water withdrawal may already exceeds the safety boundary where the demand-to-supply ratio is low (e.g. 4% for Indus), irrigating the crops in a sustainable way becomes even more challenging. Renewable ground-water has been exploited to fulfill the irrigation needs in many regions of the world, such as central North America[38], but the available renewable ground-water resources simulated by the hydrology model[38] hardly matches the above-mention regions where water deficit were large. Besides mining ground water for irrigation, the trans-basin water transfer program (e.g. the South-to-North Water Diversion Project in China) can be a viable alternative to mitigate the imbalance between water supply and demand, as the total irrigation demand over Yellow River basin and Yangtze River basin together accounts for only 1.4% of river discharge of Yangtze River.

Our analysis of the balance between irrigation demand and supply to achieve $\Delta Y$ is subject to several limitations. On the supply side, our water budget balance annually at basin scale has largely dismissed the spatial and seasonal variations of river discharge. We have also ignored hillslope constraints that may determine whether hillside croplands can use river discharge for irrigation. On the demand side, our approach likely under-estimates potential irrigation demands to close the yield gaps for two reasons. We only consider wheat and maize, while other irrigation-demanding cereals (e.g. rice), cotton, vegetable, and oil crops have not been included, due to data limitations. We estimated rainfed cropland area as the area without irrigation facilities[39], which may underestimate the area of croplands needing additional irrigation as many croplands equipped with irrigation facilities today are still rainfed or with insufficient irrigation due to economic or physical limitations[39]. Since we consider water demands from two of the many crops (lower irrigation demand) and assume all river discharge can be used for irrigation (greater irrigation supply), it should be alerting that the potential tension between irrigation demand and supply may still be underestimated. At global scale, despite growing details of spatial distribution of irrigation facilities[39], our knowledge on the amount and spatial and temporal distribution of irrigation water applied in croplands remains uncertain.

Current water constraints on closing the yield gap with additional irrigation would be exacerbated by climate change that will not only affect the size of $\Delta Y$ but also the availability of water

for irrigation[15,40]. A more explicit consideration of changes in crop yield levels, $\Delta Y$, and water availability in a common framework is thus desirable in future projections of agricultural productivity. Furthermore, closing the yield gap in countries with prevalence of undernourishment is an important contribution to food security, but its realization is often limited by "economic water scarcity" due to lack of financial capacity to build irrigation infrastructure[41]. But even for more developed countries, the economic cost of irrigation infrastructure could have been underestimated when irrigation expansion requires cross-basin water transfer for a large area.

Overall, our integrated estimate combining CA and crop models of the irrigation contribution to increase yield provides new insights for interdisciplinary studies in agronomy, hydrology, and economy. With revised $\Delta Y$ and its uncertainty estimates, our results can be used to inform policy makers where the expected yield gain is the essential factor in determining water use decisions[20]. Since global $\Delta Y$ estimated by crop models varied by a factor of 4 between models and was, on average, ~2 times larger than observations, previous hydrologic analyses rely upon one crop model or on simplified approaches could bear much biases and uncertainties in the yield difference between rainfed and irrigated yield[11,18,42]. Sustainably enhancing crop yield through irrigation could thus prove difficult. Improvement of irrigation practices could save water and allow to expand irrigation[16], as should cross-basin water transfer[43]. Without sufficient efforts on these aspects, the prospect of applying irrigation to combat adverse climate change could not be realized. As a next step, the improved estimation of $\Delta Y$ could be used for applications in the hydrology-agriculture-economic nexus[20,44,45], in which higher precisions gained for irrigation contribution to yield will propagate to better policy and decision making.

## Methods
**Attainable yield estimates from climate analogues**. The estimated rainfed and irrigated attainable yields are derived using a CA approach, which is updated from Mueller et al.[10]. A series of climatic growing zones is defined based on equal-area increments of growing degree days and precipitation. Within each climate zone, "irrigated" attainable yields are calculated as the area-weighted 95th percentile of all yield observations[33] within the bin. Specifically, the "irrigated" attainable yields were defined based on all the yield observations, including from irrigated or fractionally irrigated areas. This allows us to capture the upper end of yields for that climate zone, where farmers are using whatever management practices will allow them to maximize yields – including irrigation where it is used, particularly in drier and hotter regions. Similarly, the "rainfed" attainable yields are calculated from all yield observations within the bin that are located in a political unit with <10% of crop area irrigated, where crop-specific irrigation maps are from the

MIRCA2000 irrigation dataset[46]. The rainfed and irrigated attainable yield estimates used for this analysis are grid cell averages derived from replicating this sampling procedure for varying numbers of climate zones, from 100 to 400 ($10 \times 10$ to $20 \times 20$ growing degree day and precipitation increments, see Supplementary Fig. 8).

A limitation of this dataset is that the irrigated attainable yields may not be different from the rainfed attainable yield estimates if little area is actually irrigated within a climate zone. Further, we note that for the comparison with gridded-US data, the underlying yield dataset upon which these estimates are based[33] does include county-level USDA yield data (although these data are a combination of rainfed and irrigated observations).

**Global gridded crop models**. We used Phase 1 simulation results by global gridded crop model inter-comparison (GGCMI) results[30]. The phase 1 of GGCMI includes an unprecedented number of crop models (see Supplementary Table 2) with very different structures and assumptions. For example, photosynthesis of crops was simulated with different methods including those analogous to the Farquhar scheme and those following a light use efficiency scheme. The parameters of even identical schemes may differ across models[29]. The uncertainties of model structure and parameters were represented by this largest-ever GGCMs, which have a major difference to field-scale model applications that at large regional- to global-scale detailed model calibration to field observations is not possible[27]. Standard deviations of simulated $\Delta Y$ across the models were shown in Supplementary Fig. 9. A full list of models, their characteristics and their references can be found in Supplementary Table 2. However, all models follow the same simulation protocol[30] with the same forcing of gridded climate (AgMERRA) and management (planting date and fertilization rate) in order to minimize the impacts of difference in model drivers. Irrigation simulated by GGCMs follow the protocol with automatic irrigation that do not limit irrigation water supply[30], though different models may slightly vary in the setting, which were detailed in Muller et al.[47]. All models provided "harmnon" simulations, which simulate historical crop yield forced by historical climate dataset but assuming unlimited nutrients supply to the croplands[30] are used in the analysis. We use "harmnon" simulations with unlimited fertilizer supply because (1) it helps to avoid interactive effects of fertilization and irrigation, and (2) it is closer to the assumption used in the CA approach, making the two datasets more directly comparable. Crop rotations were not considered in current phase of GGCMI, largely due to lack of global gridded dataset of the crop rotation practices. Different GGCMs were allowed to use their own soil parameters. On the one hand, this comprises a source of uncertainties in simulated yield (e.g. Folberth et al.[29]); On the other hand, the ensemble of GGCMs with a diversity of soil parameters represents a good range of soil impacts on the crop simulations. We noted that some GGCMs also simulated other crops, including, for example, rice, rapseed, and sugarcane. However, we did not consider these due to limited number of models simulating these crops and validation datasets to evaluate model performance. In addition, flooding irrigation practices for rice over Asia has several functional features, such as facilitating transplanting. These features have not yet been well represented by GGCMs.

**Gridded US dataset**. The gridded rainfed and irrigated crop yield over US (gridded-US) dataset is based on county rainfed and irrigated yield statistics provided by the US Department of Agriculture (USDA), post-processed by Elliott et al. The statistics were gridded to 0.25° according to the weighted crop area over each county. The dataset covers 1980–2010, and we use the average across this time period, which is the common period between the field data used by CA and gridded crop model simulations. Further details of the dataset can be found in Schauberger et al.[12].

The gridded-US dataset is suitable for evaluating our reanalysis because (1) it is largely independent from both products we used in this study, and (2) unlike in many less-developed countries where rainfed farming is often paired with a significantly lower management intensity (e.g. less fertilizer input and pest control measures), the rainfed and irrigated management over the same county in US is more often associated with access to water resources. It may thus more closely approximate spatial variations in the contribution of irrigation alone to crop yield instead of the contribution of co-varying factors. To compare results from CA and GGCMs with gridded-US dataset, we decomposed mean squared deviation (MSD = $\sum_{i=1}^{n}(x_i - y_i)^2/n$, Eq. (1)) of $\Delta Y$ between CA/GGCM/the reanalysis and gridded-US dataset into the squared bias (SB = $(\bar{x} - \bar{y})^2$, Eq. (2)) and mean squared variation (MSV = $\sum_{i=1}^{n}[(x_i - \bar{x}) - (y_i - \bar{y})]^2/n$, Eq. (3)), where $\bar{x}$ and $\bar{y}$ were the means of $x_i$ and $y_i$ ($i = 1, 2...n$) respectively[48], and $x_i$ and $y_i$ in this study represented gridded $\Delta Y$ estimates from different data sources.

**River discharge**. The global river discharge dataset (UNH-GRDC) used in this study is a reanalyzed product combining observed river discharge collected by Global Runoff Data Centre (GRDC) and river discharge simulations by a hydrology model (Water Balance Model). This dataset has composite river discharge fields, which preserve the accuracy of discharge measurements as well as give the spatial and temporal patterns of it. Since the dataset has already accounted contemporary utilization of river discharge for all purposes, including irrigation, it serves in this study as an estimate of available water resources that can potentially be used to

irrigate the rainfed croplands. UNH-GRDC dataset has a spatial resolution of 0.5° and temporal resolution of 1 month over the entire global land surface[34].

River discharge data from Global River Discharge Center[37] was used as the data source for 405 river basins with mean annual discharge [km³] of the gauging station nearest to the mouth as potentially available water resources to irrigate wheat and maize croplands.

**Bayesian model average**. We derived $\Delta Y$ as the ratio of the difference between the irrigated yield and the rainfed yield to the irrigated yield (Eq. (4)).

$$\Delta Y = \frac{\text{irrigated yield} - \text{rainfed yield}}{\text{irrigated yield}} \tag{4}$$

The irrigated yield is chosen as the dividend, instead of the more intuitive rainfed yield, because the rainfed yield can be very small or even zero in extreme cases jeopardizing the stability of the analyses. The reanalysis of $\Delta Y$ integrating the global gridded crop models and climate analog approaches was performed with the Bayesian model average algorithm (BMA)[32], which has been proven to be an effective method for ensemble weather forecasts, but has not yet been applied in process-based crop model ensembles. The idea of BMA is to derive the posterior probability of each model given a target dataset. The posterior probability of each model is then used to calculate model weights ($W_i$ for the $i$th model) in the model ensemble and results in reanalyzed estimates that "best" combine information from different datasets. The derivation of $W_i$ follows the Bayesian equation (Eq. (5))

$$W_i = P(M_i|O) \propto P(O|M_i)P(M_i) \tag{5}$$

where $M_i$ is the $i$th model estimates on $\Delta Y$ and $O$ is the $\Delta Y$ estimated from the CA approach. In the prior ($P(M_i)$), we assumed each model is equally skillful in projecting $\Delta Y$. The conditional probability $P(O|M_i)$ is therefore proportional to the misfits between the $i$th model simulation and climate analog estimates. With $W_i$, the posterior probability for the best estimate of $\Delta Y$ in the reanalysis will follows Eq. (6),

$$P(\Delta Y|M_1, M_2, \ldots, M_{11}, O) = \sum_{i=1}^{11} W_i P(\Delta Y|M_i, O) \tag{6}$$

where $P(\Delta Y|M_i, O)$ is conditional probability density function of $\Delta Y$ based on $M_i$ and $O$. A Monte-carlo Markov chain method is used to derive the optimal $W_i$ for each model[32].

**Potential evapotranspiration**. In addition to temperature, we use PET as a surrogate to estimate climatic demand of water from croplands. We follow the modified Haude equation[49] to derive PET, which has been proven effective in building statistical models for regional crop yield[50]. The climatic variables used in calculating PET comes from AgMERRA dataset, which was also the climate forcing for GGCM simulations[30]. PET can also be calculated with other algorithms, such as widely used Penman–Monteith equations. However, AgMERRA does not provide the surface air pressure and downward long-wave radiation, which make deriving vapor pressure deficit and net radiation required in Penman–Monteith equations difficult. We therefore used another PET dataset, which apply Penman–Monteith equations on CRU dataset[51], as another sensitivity test to climatic water demand.

**Balance between irrigation requirements and runoff supply**. We first obtain crop water demand calculated by each GGCM, i.e., water demand without constrained by actual water availability[15,30]. To ensure consistency between reanalyzed $\Delta Y$ and water demand, we use the same model weights (see Bayesian model average) to calculate reanalyzed water demands used in the main text. The water demands estimated by GGCMs assume 100% irrigation efficiency (no conveyance or application losses), which is not ideal when comparing with available water resources. Therefore, at each grid, we divide the reanalyzed crop water demand by crop-specific irrigation efficiencies[52] to attain requested irrigation demand from surface water bodies. The crop-specific irrigation efficiency[52] is obtained from LPJmL simulation constrained by AQUASTAT data on irrigation system distribution.

On the water supply side, the UNH-GRDC dataset considers today's human water withdrawal for industrial, domestic, and agricultural usage from river runoff. River ecosystems provide life-supporting functions that depend on maintaining minimum river discharge, i.e., environmental flow requirements (EFRs)[53,54]. However, the quantification of EFRs is not trivial as estimation methods vary, leading to a broad range of uncertainty (e.g. EFRs estimated by different methods ranges from 12 to 48% for Nile and 30 to 67% for the Amazon). A detailed account of such is beyond the scope of this study and can be found in Jägermeyr et al.[11]. The large uncertainties in EFRs and lack of data for other potential water usages make it difficult to estimate the overall sustainable water resources available for irrigating wheat and maize. In this study we therefore assume a conservative, but static fraction of river discharge accessible for irrigating, set to 20–40%, which covers the typical range of water withdraw considered in previous studies[11,15]. The choice of 20% or 40% will not qualitatively change our findings (Supplementary Fig. 5). Confirming our assumptions, Elliott et al.[15] assumed that up to 40% of naturalized runoff might be used human needs including irrigation. We do not consider renewable groundwater, which could provide additional source of irrigation water. With a rough estimate from a recent modeling study by Rosegrant and Kai[42] at global scale, renewable groundwater use for

irrigation is ~383 km$^3$/year, which is about one-fifth of the total surface water use (1993 km$^3$/year) for irrigation[42]. But the potential to fulfill future irrigation needs are limited since the resources are not matching with the regions where water deficit were large. Given that both current human water withdrawals are already unsustainable across many river basins worldwide[11], that there are many other irrigation-intensive crops in addition to the here-studied wheat and maize (e.g. rice, cotton, and vegetables), and that the seasonality of river discharge, flood, and droughts were not considered in this multi-year average balance accounting, the estimates of water deficit in our study appear conservative.

**Reporting summary**. Further information on research design is available in the Nature Research Reporting Summary linked to this article.

## Data availability

GGCMI model data can be accessed under open data license CC BY 4.0 in https://zenodo.org (Supplementary Table 4). Attainable irrigated and rainfed yield by Climate Analog approach can be accessed upon request to Prof. N Mueller (nathan.mueller@rams.colostate.edu). Global gridded river discharge can be downloaded from http://www.grdc.sr.unh.edu/.

## Code availability

The analyses were performed using MATLAB (R2017b). Computer codes for the analyses are available upon request to the corresponding author.

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

## Acknowledgements

This study was supported by the National Key R&D Program of China (2017YFA0604702 & 2019YFA0607302) and National Natural Science Foundation of China (42041007 & 41988101).

## Author contributions

X.W. designed the study and performed the analyses. X.W., C.M., and N.D.M. wrote the manuscript. X.W., C.M., J.E., D.D., C.F., W.L., S.O., T.A.M.P., A.R., and E.S. performed global crop model simulations. N.M. provided climate analogue dataset. All authors contributed to interpretation of the results and manuscript revisions.

## Competing interests

The authors declare no competing interests.
