## [Peer Review File · Nature Communications]

Reviewers Comments, first round -

Reviewer #1 (Remarks to the Author):

Review of paper „global irrigation contribution to wheat and maize yield” by Wang et al.

General comments

The paper integrates empirical estimates with simulations of gridded global crop models to explore yield gaps between rainfed and irrigated conditions.

The topic is interesting, relevant and timely. The methodology is an improvement compared to widely spread model-based approaches or empirical studies, and single-model studies. The paper is well written and has a clear structure. The study mentions some limitations of the methodology, which is laudable.

However, I think this work would benefit from adding some additional issues (see specific comments).

Specific comments

I wonder why considering only wheat and maize. While both of them are important for global food security, rice plays also a major role in Asia and should, in my opinion, be included in the study (also because of its high water use).

Renewable water resources are represented by river streamflow. This seems to be very limiting for some areas with large use of groundwater. While fossil groundwater may be left out of the scope due to its unsustainable character, I do not find any good argument for excluding renewable groundwater. Of course, data is essentially hard to get, but not impossible to roughly be simulated by some models.

CA method: please better explain how this considers or neglects soil heterogeneity and geomorphology within the same climate zone.

Please explain why GGCMs have a systematic overestimation compared to USDA data.

L334 “[...] static fraction of river discharge accessible for irrigating, set to 20-40%” please clarify, is it 20 % in some basins and 40 in others? Is it two simulations, one with 20, one with 40?

Figure 2: I assume delta Y is displayed here using the BMA method, please state in the caption. I would find much more valuable and useful that you show the uncertainty range not for a latitudinal band but per region, for example as shadow above the colors or bars or something else.

Caption of Fig 2: “shaded area indicates the range of uncertainty”. Is it the standard deviation? Is it the uncertainty coming from the difference in model results? Please clarify. Also please include the methodology to assess uncertainty in the Method section. And also please discuss a bit more what this uncertainty means for the use of your results and attach uncertainty information to the description of your results. For example, in what regions is the uncertainty low enough to be useful for stakeholders?

L 211 “The reanalyzed ΔY can be used directly to inform local irrigation needs when the expected yield gain is the essential factor in determining water use decisions¹⁷” I do not quite agree with this statement. How should a stakeholder use your results? Should they try to see in your maps what color the legends are and what all those technical terms mean? Should they look at Table 1 and assume that these values are constant all over a certain river basin? I suggest improving the relevance and usability of the study by compiling a very simple interactive map where

stakeholders can click and see the range of delta Y with uncertainty information, accompanied by a short, simple and understandable paragraph of how this interactive map was made and for what it can and can't be used. Otherwise this is a purely academic exercise without practical use outside science.

Extended data Fig 1: "mean growing season precipitation" mean over many years? Mean over months? Please clarify.

Minor comments

L52 "[...] precipitation than evaporative demand [...]", do you mean "[...]precipitation and evaporative demand[...]"?

L64 "minimizes". Shouldn't it be "minimize"?

L65 "[...] which is expected to result in substantial net declines in regional to global crop yields" This statement is in my view too simple and does not make justice to the debate on CO₂-fertilization, spatial heterogeneity of responses, or differences between crops. Please reformulate.

L219 "[...] to for [...]" Please check.

Caption Fig 1., replace "yaxis" by "y-axis"

Reviewer #2 (Remarks to the Author):

Key results: A global scope of yield gains associated with irrigated vs rainfed production for two major crops is very interesting and can help guide agricultural policy at local and aggregate levels. There are also several interesting aspects of the technical analysis: (i) multiple datasets are integrated in a coherent way, (ii) both statistical and process-based models are considered, and (iii) a Bayesian approach is used to combined insights from both. The magnitude of the average (global) yield gains associated with irrigation were not surprising, but the range of localized (cross-sectional) gains was.

Validity: I have three main concerns: fit for the journal, measurement of a key variable, and validity of the BMA model.

1. The NCC states on its Aims and Scope webpage that "[NCC] publishes cutting-edge research on the science of contemporary climate change, its impacts, and the wider implications for the economy, society and policy. " In the manuscript, climate change is used to motivate the research (first P) but the analysis itself does not provide any information on climate change impacts. Thus, the paper does not seem a good fit for the journal unless it leverages the models to predict climate change impacts. I think a 1/2C warming temp simulation (holding precip fixed) would be very interesting as (presumably) the benefit of irrigation becomes larger under warming temps (in some/most areas).

2. I was confused on how exactly irrigated yields were defined/measured. In the methods section the authors note that "Within each growing zone, "irrigated" attainable yields are calculated as the area-weighted 95th percentile of all yield observations within the bin." My literal reading of this sentence suggests to me that for a certain unit, the authors pooled all the yield observations - which could include both rainfed and irrigated - and then identified the 95th percentile. Or did they use the 95th percentile of just the irrigated observations? Can the authors please provide more information on the measurement of this key variable? Maybe a few specific explanations for the following cases would be helpful:

- i. a bin in which there is currently no irrigation (BTW do these exist in the data?)
- ii. a bin that has both irrigated and rainfed land
- iii. a bin that only has irrigation (exist in data?)

3. The third column of figure 1 suggests that there is still a non-trivial bias associated with with the BMA model? Maybe I missed it, but was a cross-validation approach considered? It would seem informative to consider training the models on a (random) subset of locations, and then predicting Delta_y on a portion that are left out. This could be done for all three models for comparison.

Originality and significance: The manuscript does not contain any clear statement on the novelty of its methods, data, nor conclusions. To me, no one individual part of the data/methods is novel, but perhaps the integration of them is? I do think the results are interesting for multiple disciplines, but it is not clear how knowledge is being expanded here. What are the new insights?

Data & methodology: The data and methods are solid for research with a global scope, outside of concerns raised above. The authors do a nice job of identifying and discussing limitations. Its not clear to me how parameter uncertainty of the underlying models (both statistical and process based) is being captured in the reported results, more clarification is need on that.

Conclusions: The conclusions are reasonable given the analysis, but again the reader is not gaining any insights into climate change implications.

References: The manuscript can be enhanced by including cites to relevant literature.

1. The first paper I saw that combined stat and process models was:

Roberts, Michael J., et al. "Comparing and combining process-based crop models and statistical models with some implications for climate change." *Environmental Research Letters* 12.9 (2017): 095010.

The current manuscript does not discuss the currently evolving literature that combines stat and process models. It pulls up short of claiming novelty, which is appropriate, but it would be good to know if the BMA approach is a contribution to this literature. Essentially, a couple sentences on what has been done before and what is novel.

2. The recently evolving literature on climate/irrigation interactions can be used to help motivate the analysis, some papers include:

Li, X., and T. J. Troy. "Changes in rainfed and irrigated crop yield response to climate in the western US." *Environmental Research Letters* 13.6 (2018): 064031.

Tack, Jesse, Andrew Barkley, and Nathan Hendricks. "Irrigation offsets wheat yield reductions from warming temperatures." *Environmental Research Letters* 12.11 (2017): 114027.

Troy, Tara J., Chinpihoi Kipgen, and Indrani Pal. "The impact of climate extremes and irrigation on US crop yields." *Environmental Research Letters* 10.5 (2015): 054013.

Carter, Elizabeth K., et al. "Separating heat stress from moisture stress: analyzing yield response to high temperature in irrigated maize." *Environmental Research Letters* 11.9 (2016): 094012.

Clarity and context: The manuscript is well written and clear.

Reviewer #3 (Remarks to the Author):

This manuscript describes a combined approach to assess the potential irrigation contribution in closing yield gap of two important staple crops: wheat and Maize. The use of combined approach climate analogues and multi-crop modelling methods to assess the irrigation potential to reduce yield gaps is quite interesting and the advantages from both methods are well described. Though the complexity of the farming system and water contribution to productivity were largely simplified in this work but this is expected given the global scale of this study.

The two selected crops have probably the largest number of varieties commercially grown across the globe (some are GMO) and each has different growing seasons, different levels of adaptation

to climate uncertainties, water stress, salinity... Hence different level of productivity and water demand even under "the same climate" are simply related to variety selection, nutrient management, physical and chemical soil characteristics... and not necessarily to irrigation as authors stated several times in this paper. Also the original studies of yield "survey" data, climatic zones and USDA yield dataset have each their own limitations and assumptions and hence the results presented in this paper must be carefully interpreted and authors should stress more on the limitations of this study.

Again, given the scale of the study many of those assumptions can be accepted but also must be clearly presented to the readers. Therefore, I would suggest authors to go more in depth on the discussion/methodological limitation.

Specific comments

Line 51: how can the variability 23% be more than the average yield change (22%) for maize? This suggests that irrigation might have zero to negative effect on yield which is not logical otherwise why bother to irrigate. Please adjust.

Line 56: Would you be able to quantify the volumetric need of water to close the yield gap?

Line 66: expanding irrigation do not achieve higher yield levels or improve resiliency. It only improve production not productivity, please remove "expanding" or adjust the sentence.

Line 71-73: What you mean by one or two coefficient of ET for each crop? If authors are referring to crop coefficients K_c used to adjust reference ET (one = single K_c and two = dual k_c) then there is misconception of the entire crop coefficient approach. K_c is an adjustment factor of the climatic parameter (grass/reference evapotranspiration (E_{To})) and relates to the crop development stages and plant status. Hence K_c has nothing to do with climatic variability. Please remove or adjust this sentence as it is conceptually wrong. Also regarding assumptions and limitations: Most K_c available in literature were produced from drainage lysimeters or fully irrigated experiments and hence do not capture deficit irrigation which is highly practiced especially on the two crops analysed in this manuscript.

Line 80: Please provide a brief description of your survey and census data used in Manfreda et al. Where they come from? They correspond to what year? are they actually survey and census data? Do they include irrigated and rainfed data? What happen if the survey data do not correspond to an average year (i.e wet year or dry year)? Rainfed yield can be highly sensitive to rainfall especially in humid climate or crops relying on winter rain such as the two crops studied here.

Line 80: The climatic zones maps is based on which year? Single year/historical climate...? What are the criteria used? Climatic zones might differ from one year to another and can also be specific to crops. What about soil parameters should not also these be taken into consideration?

Line 94: Crop models are biophysical models and actually have the ability to account for crop varieties, management practices and soil properties. The limitation is not in the model capabilities but in the lack of data to properly operate those models? Even if the study is taken from another previously published study, authors must provide brief description of the assumptions and parameters used for simulation. For example, those multi-models used were calibrated based on winter or summer wheat? What type of irrigation management was used, full or supplemental irrigation? Are all those models been calibrated on the same dataset?

Line 113: why the GGCM clearly over-estimate dY ? Is that because those models assume full irrigation? Supplemental irrigation is highly practiced on low value crops such as cereals and mainly wheat.

Line 118: coefficient of correlation r is not enough to judge a good or bad result. You can have good prediction with low r so I would suggest to add RMSE to your analysis too.

Line 144: There is also a temporal variation which could be as much important as the spatial one

Line 145: precipitation and PET used in those correlations are based on 12 month values or on the months of the crop growing season? If are based on 12 month, I would suggest to use precipitation and PET per growing season instead. I suggest to use more complex climatic indicators that take into account both precipitation and ET for this kind of correlation.

Line 150-152: what you mean by PET and rainfall variability? Is this spatial variability within each agro-climatic zone or temporal variability per grid? if it is temporal what time period was used?

Line 156: again same as before "expanding" can't reduce yield gap

Line 157: The volumetric water demand to reduce the gap is much more interesting than yield gap correlation with PET and rainfall. I would suggest to add volumetric water demand maps or table if possible.

Line 158: Using the global river discharge to assess water availability to reduce yield gap is not

much relevant for two crops such as wheat and maize. Surface flow might not be available but what about ground water? Also, if water is made available I strongly doubt that it will go to wheat and Maize rather to more cash crops that can justify the extra costs of irrigation investment. Also assuming 20 to 40% of river discharge is used for irrigation is too crude. Instead I would suggest to link water demand to other water risk indicators such as the one produced by the aqueduct project and include baseline water stress, interannual variability, seasonal variability, flood occurrence, and drought severity.

Line 216: remove "to"

Line 224: explain better what increments of GDD means. Also can you add a map of the climatic zones of Mueller et al. used in this work?

Line 227: Political unit or climatic zone?

Line 228: Irrigated map used corresponds to which year? Rainfed wheat is largely used in rotation with other irrigated crops (i.e vegetables) in many parts of the world.

Line 230: what you mean by 100 to 400? Have you used two climatic zones maps? One for GDD and one for precipitation? Please explain more, also the word increments too.

Line 243: Explain in two sentences what parameters were used to calibrate those models and where those data were taken from?

Line 247: What is harmnon and full harm simulation settings are very specific. I would suggest to limit the description to the unlimited nutrient supply

Line 258: Instead of using the average across 1980-2010, would it be possible to limit your analysis to a single year i.e. 2010? Doing so, you would be consistent with the data produced by climatic zones, survey data and USDA dataset.

Line 271-272: Rephrase

Line 315: replace "potential irrigation water withdraw" with "crop water demand" or "irrigation water requirement"

Line 322: what is the crop specific irrigation efficiencies? The actual irrigation withdrawals is the one measured using flow meters which is not the case here.

Reviewer #1

[Reviewer 1 Comment 1] *The paper integrates empirical estimates with simulations of gridded global crop models to explore yield gaps between rainfed and irrigated conditions.*

The topic is interesting, relevant and timely. The methodology is an improvement compared to widely spread model-based approaches or empirical studies, and single-model studies. The paper is well written and has a clear structure. The study mentions some limitations of the methodology, which is laudable.

However, I think this work would benefit from adding some additional issues (see specific comments).

[Response] We thank the reviewer for constructive evaluations and suggestions. We have accordingly revised our manuscript following the reviewer's suggestion, with point-by-point response below.

Specific comments

[Reviewer 1 Comment 2] *I wonder why considering only wheat and maize. While both of them are important for global food security, rice plays also a major role in Asia and should, in my opinion, be included in the study (also because of its high water use).*

[Response] We agree with the reviewer that rice is an important staple food for a large population mainly in Asia. However, we do not include rice in this study for two considerations: 1) rice is often irrigated with flooding, the flooding irrigation not only supplies water but also becomes a management measure to control number of tillers and the transition between vegetative and reproductive growth (e.g. Ginigaddara & Ranamukhaarachchi, 2016), which is a widespread practice at least over China and Thailand. 2) rainfed lowland rice, which is common in Southeast Asia countries, is generally categorized as rainfed, but its water supply depends heavily on local landscape. Sometimes, this "rainfed" type may experience more water supply than irrigated rice. In this case, the yield difference between irrigated and rainfed rice is in a quite different concept than wheat and maize.

Both the climate analogue approach and many of the gridded crop models have not yet

developed the capacity to deal with the above-mentioned complexity of rice irrigation. Therefore, we do not include rice in this study. But under the framework of global gridded crop model intercomparison project, we are developing the models on this aspect and going to inter-compare and evaluate the model performance for rice in future studies. Thanks for your understanding.

[Reviewer 1 Comment 3] *Renewable water resources are represented by river streamflow. This seems to be very limiting for some areas with large use of groundwater. While fossil groundwater may be left out of the scope due to its unsustainable character, I do not find any good argument for excluding renewable groundwater. Of course, data is essentially hard to get, but not impossible to roughly be simulated by some models.*

[Response] We use an observational river discharge data set to constrain irrigation water use per grid cell and per basin. This assumption excludes fossil groundwater use, which is under the assumption that expanding irrigation should not be done in an unsustainable manner. Within larger river basins, renewable groundwater often feeds river streamflow further downstream so that this part of renewable groundwater should be partly reflected in the discharge data set. Thus, renewable groundwater was not fully excluded from our analyses. However, we agree with the reviewer that the issue is complex and should be mentioned explicitly in the manuscript. Hydrological models may be able to distinguish surface water from renewable groundwater discharge, but in fact many do not do this distinction and differ substantially in their estimates of river discharge (Schewe et al. 2014). In the revised manuscript we followed this suggestion by better discussing these assumptions and implications for underestimating the renewable water resources in regions where renewable groundwater could provide additional source of irrigation water with a rough estimate from a recent modelling study by Hanasaki et al. (2018). According to H08 hydrology model simulation, at global scale, renewable groundwater use for irrigation is about 383 km³/year, which is about one-fifth of the total surface water use (1993 km³/year) for irrigation (Hanasaki et al., 2018). But the potential to fulfill future irrigation needs are limited since the resources are not matching with the regions where water deficit were large.

[Reviewer 1 Comment 4] *CA method: please better explain how this considers or neglects soil heterogeneity and geomorphology within the same climate zone.*

[Response] The climate analog method neglects soil heterogeneity and geomorphology within the same climate zone, and is a limitation of the method as compared to the crop models. We have made this more explicit in the manuscript on lines 87 and 93-95.

[Reviewer 1 Comment 5] *Please explain why GGCMs have a systematic overestimation compared to USDA data.*

[Response] Thanks for pointing it out. The GGCM protocol requested all models to perform automatic irrigation scheme triggering irrigation with little moisture stress and unlimited amount of irrigation water supply (Elliott et al., 2015). Although different models vary in their representation (Muller et al., 2019), this setting leads to more irrigation application than the reality in the US and most other places in the world. This highlights the need to further improve model representation of irrigation and modelling protocol to apply irrigation in the future. We have accordingly included this information in the revised Methods section.

[Reviewer 1 Comment 6] *L334 “[...] static fraction of river discharge accessible for irrigating, set to 20-40%” please clarify, is it 20 % in some basins and 40 in others? Is it two simulations, one with 20, one with 40?*

[Response] We followed this suggestion by clarifying that 20-40% represents the range of reasonable percentage of river discharge that could be used in irrigation application, which is a range represents typical assumptions of river discharge used for irrigation (e.g. Elliot et al., 2014; Jägermeyr et al., 2017). They are represented in 21 simulations which sampling the range by 1% equal interval.

[Reviewer 1 Comment 7] *Figure 2: I assume delta Y is displayed here using the BMA method, please state in the caption. I would find much more valuable and useful that you show the uncertainty range not for a latitudinal band but per region, for example as shadow above the colors or bars or something else.*

[Response] We have accordingly revised captions of Figure 2, and provided a supplementary

Table including for each major		Wheat		Maize		reanalyzed ΔY crop producers.
		China	42.2	USA	24.9	
	India	53.5	China	22.6		
Table R1.	Russia	15.7	Brazil	22.2		irrigation yield (ΔY) (%)
	USA	31.9	France	24.4		

contribution to
for major wheat and maize producers

[Reviewer 1 Comment 8] *Caption of Fig 2: “shaded area indicates the range of uncertainty”. Is it the standard deviation? Is it the uncertainty coming from the difference in model results? Please clarify. Also please include the methodology to assess uncertainty in the Method section. And also please discuss a bit more what this uncertainty means for the use of your results and attach uncertainty information to the description of your results. For example, in what regions is the uncertainty low enough to be useful for stakeholders?*

[Response] Following this suggestion, we have clarified the shaded area in the caption of Figure 2 as the model spread. We have accordingly revised the uncertainty assessment and detailed its description in the Methods section. We further provided maps of inter-model spread (Supplementary Figure 7) following the suggestion.

[Reviewer 1 Comment 9] *L 211 “The reanalyzed ΔY can be used directly to inform local irrigation needs when the expected yield gain is the essential factor in determining water use decisions¹⁷” I do not quite agree with this statement. How should a stakeholder use your results? Should they try to see in your maps what color the legends are and what all those technical terms mean? Should they look at Table 1 and assume that these values are constant all over a certain river basin? I suggest improving the relevance and usability of the study by compiling a very simple interactive map where stakeholders can click and see the range of delta Y with uncertainty information, accompanied by a short, simple and understandable*

paragraph of how this interactive map was made and for what it can and can't be used. Otherwise this is a purely academic exercise without practical use outside science.

[Response] Following this suggestion, we have been in contact with an IT company to provide online interactive access of our products, with a prominent feature that the user will be able to select the scale of their interest (e.g. a grid point, a sub-basin or a country/region). With this tool they could easily obtain the information for the policy decision. Due to the COVID-19 pandemic, the process is delayed. We thank the reviewer for the suggestion and we expect the IT effort as a separate work to be achieved in the short future. This does not affect the value of our results that are interesting not only to the scientific community but also to the policy makers. In addition, we provided the country specific estimates for major wheat and maize producers as a supplementary table.

[Reviewer 1 Comment 10] *Extended data Fig 1: “mean growing season precipitation” mean over many years? Mean over months? Please clarify.*

[Response] We have accordingly revised the figure caption, clarifying that is the mean over 1980-2010.

Minor comments

[Reviewer 1 Comment 11] *L52 “[...] precipitation than evaporative demand [...]”, do you mean “[...]precipitation and evaporative demand[...]*”?

[Response] It is accordingly clarified as “more driven by regional patterns of precipitation than evaporative demand.”

[Reviewer 1 Comment 12] *L64 “minimizes”. Shouldn't it be “minimize”?*

[Response] Corrected accordingly.

[Reviewer 1 Comment 13] *L65 “[...] which is expected to result in substantial net declines in regional to global crop yields” This statement is in my view too simple and does not make justice to the debate on CO₂-fertilization, spatial heterogeneity of responses, or differences between crops. Please reformulate.*

[Response] We agree with the reviewer that “substantial net declines” was oversimplifying the complexity associated with climate change and removed this statement accordingly.

[Reviewer 1 Comment 14] L219 “[...] to for [...]” Please check.

[Response] “to” removed accordingly.

[Reviewer 1 Comment 15] *Caption Fig 1., replace “yaxis” by “y-axis”*

[Response] Corrected accordingly.

Reviewer #2

[Reviewer 2 Comment 1] *Key results: A global scope of yield gains associated with irrigated vs rainfed production for two major crops is very interesting and can help guide agricultural policy at local and aggregate levels. There are also several interesting aspects of the technical analysis: (i) multiple datasets are integrated in a coherent way, (ii) both statistical and process-based models are considered, and (iii) a Bayesian approach is used to combined insights from both. The magnitude of the average (global) yield gains associated with irrigation were not surprising, but the range of localized (cross-sectional) gains was.*

Validity: I have three main concerns: fit for the journal, measurement of a key variable, and validity of the BMA model.

[Response] We thank the reviewer for the constructive review and suggestions. Point-by-point response was provided below.

[Reviewer 2 Comment 2] *1. The NCC states on its Aims and Scope webpage that "[NCC] publishes cutting-edge research on the science of contemporary climate change, its impacts, and the wider implications for the economy, society and policy. " In the manuscript, climate change is used to motivate the research (first P) but the analysis itself does not provide any information on climate change impacts. Thus, the paper does not seem a good fit for the journal unless it leverages the models to predict climate change impacts. I think a 1/2C warming temp simulation (holding precip fixed) would be very interesting as (presumably) the benefit of irrigation becomes larger under warming temps (in some/most areas).*

[Response] The paper is now transferred to *Nature Communications*, which we believe is a good fit to the journal due to its impacts and inter-disciplinary readership.

[Reviewer 2 Comment 3] *2. I was confused on how exactly irrigated yields were defined/measured. In the methods section the authors note that "Within each growing zone, "irrigated" attainable yields are calculated as the area-weighted 95th percentile of all yield observations within the bin." My literal reading of this sentence suggests to me that for a certain unit, the authors pooled all the yield observations - which could include both rainfed*

and irrigated - and then identified the 95th percentile. Or did they use the 95th percentile of just the irrigated observations? Can the authors please provide more information on the measurement of this key variable? Maybe a few specific explanations for the following cases would be helpful:

i. a bin in which there is currently no irrigation (BTW do these exist in the data?)

ii. a bin that has both irrigated and rainfed land

iii. a bin that only has irrigation (exist in data?)

[Response] The reviewer is correct – the sentence is meant to be interpreted literally. Following the reviewer’s suggestion, we have further clarified the methods used in CA approach. We first defined the “irrigated” attainable yields using all the yield observations, including from irrigated or fractionally irrigated areas. This allows us to capture the upper end of yields for that climate zone, where farmers are using whatever management practices will allow them to maximize yields – including irrigation where it is used, particularly in drier and hotter regions. We then explicitly remove irrigated observations to calculate the rainfed attainable yields. Because the true native resolution of the yield observations are political units, we aggregate irrigation statistics up to political units, and then proceed to exclude all political units where more than 10% of harvested area is irrigated.

For maize, we illustrate below the three examples requested by the reviewer for the set of 10x10 (GDD x precipitation) climate bins. Each plot shows the cumulative distribution of yields (y-axis) across harvested areas (x-axis). The 95th percentile of hectares (indicated by the thin lines) provides the estimate of attainable yield.

- i. A bin in which there is very little irrigation (11.6% of harvested area is classified as rainfed using our approach):

ii. A bin that has approximately equal irrigated and rainfed land:

iii. A bin that has a majority of irrigated area (77.4% of harvested area is classified as irrigated). One can see in this plot that almost all of the high-yielding areas are irrigated for this bin:

[Reviewer 2 Comment 4] 3. *The third column of figure 1 suggests that there is still a non-trivial bias associated with the BMA model? Maybe I missed it, but was a cross-validation approach considered? It would seem informative to consider training the models on a (random) subset of locations, and then predicting Δy on a portion that are left out. This could be done for all three models for comparison.*

[Response] We have clarified in the revised manuscript that what Figure 1 shows was not for training, but as a third-party independent dataset to illustrate the strength and weakness of the CA, GGCM and the reanalysis dataset. Cross-validation was used to test the BMA, with the resulting ΔY vary from the complete reanalysis by less than 1%, indicating our results were quite robust to the spatial variations in ΔY .

[Reviewer 2 Comment 5] *Originality and significance: The manuscript does not contain any clear statement on the novelty of its methods, data, nor conclusions. To me, no one individual part of the data/methods is novel, but perhaps the integration of them is? I do think the results are interesting for multiple disciplines, but it is not clear how knowledge is being expanded here. What are the new insights?*

[Response] The primary novelty in our methodology is developing the framework to integrate and reconcile the two diverging approaches estimating contribution of irrigation to crop yield, providing a consistent global estimate of ΔY , without making any assumption on its drivers. We also provided reasonable uncertainty estimates to be used in multiple disciplines. As demonstrated by Figure 1 and the associated statistics, the performance of our reanalysis outweighs either approach with low absolute bias and retain good spatial coherence. As the other reviewer noted, it is a clear improvement compared to widely spread model-based approaches or empirical studies, and single-model studies. We have accordingly clarified it in the revised manuscript. In addition, we did exploratory efforts to calculate how our reanalysis implies for the potentials in expanding irrigation practice with renewable water resources, which has never been done before due to large uncertainties in crop irrigation demand. Thanks for the suggestion!

[Reviewer 2 Comment 6] *Data & methodology: The data and methods are solid for research with a global scope, outside of concerns raised above. The authors do a nice job of identifying and discussing limitations. Its not clear to me how parameter uncertainty of the underlying models (both statistical and process based) is being captured in the reported results, more clarification is need on that.*

[Response] The structure and parameter uncertainties of GGCMs were represented by the ensemble of 10 models, we do not specifically test sensitivity to a specific parameter in these process models. In the revised Methods section, we have further extended descriptions on the CA approach and its uncertainty estimates. In addition, following Comment 4 of this reviewer, we have revised the description of uncertainties consistently to represent inter-model spread, which has further helped us identifying the hotspot of future researches. Thanks for the constructive suggestion!

[Reviewer 2 Comment 7] *Conclusions: The conclusions are reasonable given the analysis, but again the reader is not gaining any insights into climate change implications.*

[Response] Following the suggestion, we have revised the conclusion to further expand the conclusion for how to sustainably enhancing crop production with irrigation, as well as highlighting the usefulness of our framework for the application in irrigation related policy making.

[Reviewer 2 Comment 8] *References: The manuscript can be enhanced by including cites to relevant literature.*

1. The first paper I saw that combined stat and process models was:

Roberts, Michael J., et al. "Comparing and combining process-based crop models and statistical models with some implications for climate change." Environmental Research Letters 12.9 (2017): 095010.

The current manuscript does not discuss the currently evolving literature that combines stat

and process models. It pulls up short of claiming novelty, which is appropriate, but it would be good to know if the BMA approach is a contribution to this literature. Essentially, a couple sentences on what has been done before and what is novel.

2. The recently evolving literature on climate/irrigation interactions can be used to help motivate the analysis, some papers include:

*Li, X., and T. J. Troy. "Changes in rainfed and irrigated crop yield response to climate in the western US." *Environmental Research Letters* 13.6 (2018): 064031.*

*Tack, Jesse, Andrew Barkley, and Nathan Hendricks. "Irrigation offsets wheat yield reductions from warming temperatures." *Environmental Research Letters* 12.11 (2017): 114027.*

*Troy, Tara J., Chinpihoi Kipgen, and Indrani Pal. "The impact of climate extremes and irrigation on US crop yields." *Environmental Research Letters* 10.5 (2015): 054013.*

*Carter, Elizabeth K., et al. "Separating heat stress from moisture stress: analyzing yield response to high temperature in irrigated maize." *Environmental Research Letters* 11.9 (2016): 094012.*

[Response] Following the reviewer's suggestion, we have included the recommended references in the revised manuscript. The Introduction is improved through putting these efforts in context, which help highlight our improvements made in integrating process and statistical models in the methodological aspect. Discussions were also improved with the provided references. Thanks for the suggestion!

[Reviewer 2 Comment 9] *Clarity and context: The manuscript is well written and clear.*

[Response] Thanks!

Reviewer #3

[Reviewer 3 Comment 1] *This manuscript describes a combined approach to assess the potential irrigation contribution in closing yield gap of two important staple crops: wheat and Maize. The use of combined approach climate analogues and multi-crop modelling methods to assess the irrigation potential to reduce yield gaps is quite interesting and the advantages from both methods are well described. Though the complexity of the farming system and water contribution to productivity were largely simplified in this work but this is expected given the global scale of this study.*

The two selected crops have probably the largest number of varieties commercially grown across the globe (some are GMO) and each has different growing seasons, different levels of adaptation to climate uncertainties, water stress, salinity... Hence different level of productivity and water demand even under “the same climate” are simply related to variety selection, nutrient management, physical and chemical soil characteristics... and not necessarily to irrigation as authors stated several times in this paper. Also the original studies of yield “survey” data, climatic zones and USDA yield dataset have each their own limitations and assumptions and hence the results presented in this paper must be carefully interpreted and authors should stress more on the limitations of this study.

Again, given the scale of the study many of those assumptions can be accepted but also must be clearly presented to the readers. Therefore, I would suggest authors to go more in depth on the discussion/methodological limitation.

[Response] We thank the reviewer for the constructive evaluation and suggestions. Following the suggestions, we have carefully revised the manuscript with further in-depth discussions on the limitations of the methodologies. We appreciated the reviewer acknowledging that at global scale, this represents what could be achieved with contemporary information and we have also indicated the ways forward to further improve the assessments in future studies.

Specific comments

[Reviewer 3 Comment 2] *Line 51: how can the variability 23% be more than the average yield change (22%) for maize? This suggests that irrigation might have zero to negative effect*

on yield which is not logical otherwise why bother to irrigate. Please adjust.

[Response] 23% represents spatial variations in ΔY , which can go beyond its mean, but the reviewer is correct that it may not truly represent the uncertainties of ΔY estimates. Following this suggestion, we have revised the description of uncertainties to represent the model spread consistently throughout the manuscript.

[Reviewer 3 Comment 2] *Line 56: Would you be able to quantify the volumetric need of water to close the yield gap?*

[Response] Following this suggestion, we have provided the volumetric need of irrigation water at river basin scale in Supplementary Table 1 and further discussed the uncertainties and implications of irrigation water requirements in the revised manuscript. Thanks for the suggestion.

[Reviewer 3 Comment 3] *Line 66: expanding irrigation do not achieve higher yield levels or improve resiliency. It only improve production not productivity, please remove “expanding” or adjust the sentence.*

[Response] “expanding” removed accordingly.

[Reviewer 3 Comment 4] *Line 71-73: What you mean by one or two coefficient of ET for each crop? If authors are referring to crop coefficients K_c used to adjust reference ET (one = single K_c and two = dual k_c) then there is misconception of the entire crop coefficient approach. K_c is an adjustment factor of the climatic parameter (grass/reference evapotranspiration (E_{To})) and relates to the crop development stages and plant status. Hence K_c has nothing to do with climatic variability. Please remove or adjust this sentence as it is conceptually wrong. Also regarding assumptions and limitations: Most K_c available in literature were produced from drainage lysimeters or fully irrigated experiments and hence do not capture deficit irrigation which is highly practiced especially on the two crops analysed in this manuscript.*

[Response] Following this suggestion, we have removed the discussions on crop coefficients

in this paragraph. We have extended the discussion on how crop modelling of irrigation water demand could improve from empirical determination of crop coefficients (K_c) to more process-oriented modelling in the Discussion section. Thanks for the suggestion!

[Reviewer 3 Comment 5] *Line 80: Please provide a brief description of your survey and census data used in Manfreda et al. Where they come from? They correspond to what year? are they actually survey and census data? Do they include irrigated and rainfed data? What happen if the survey data do not correspond to an average year (i.e wet year or dry year)? Rainfed yield can be highly sensitive to rainfall especially in humid climate or crops relying on winter rain such as the two crops studied here.*

[Response] Thanks for the suggestion. The survey and census data used here were not from Manfreda et al., but from Mueller et al. (2012). The details were provided in the reference and Methods section. We have accordingly further clarified the dataset used in the revised manuscript on lines 87 and 93-95. The calculation of attainable rainfed and irrigated yield were further detailed in the revised Methods section with supplementary figure (Supplementary Figure 6).

[Reviewer 3 Comment 6] *Line 80: The climatic zones maps is based on which year? Single year/historical climate...? What are the criteria used? Climatic zones might differ from one year to another and can also be specific to crops. What about soil parameters should not also these be taken into consideration?*

[Response] The climatic zone maps are crop-specific multi-year average (Mueller et al., 2012), which is now clarified in the revised Methods section with a supplementary figure. They are used according to the range of field data collected. The climatic zones used here are indeed crop specific (Supplementary Figure 6). We also agree with the reviewer that soil parameters could be a source of uncertainties (e.g. Folberth et al., 2016). Although GGCMs are not able to be harmonized for the soil parameters (Muller et al., 2019), the diversity of soil parameters used by GGCMs may better reflect the soil impacts on crop simulations than a single harmonized assumption. In the revised manuscript, have clarified this aspect in the Methods section.

[Reviewer 3 Comment 7] *Line 94: Crop models are biophysical models and actually have the ability to account for crop varieties, management practices and soil properties. The limitation is not in the model capabilities but in the lack of data to properly operate those models? Even if the study is taken from another previously published study, authors must provide brief description of the assumptions and parameters used for simulation. For example, those multi-models used were calibrated based on winter or summer wheat? What type of irrigation management was used, full or supplemental irrigation? Are all those models been calibrated on the same dataset?*

[Response] We agree with the reviewer that the lack of full consideration for crop varieties, management practices and soil properties is largely due to lack of proper data to force the model and accordingly revise the statements. We have also followed this suggestion to further describe the model simulations in detail in the revised Methods section.

[Reviewer 3 Comment 8] *Line 113: why the GGCM clearly over-estimate dY ? Is that because those models assume full irrigation? Supplemental irrigation is highly practiced on low value crops such as cereals and mainly wheat.*

[Response] Thanks for pointing it out. The GGCM protocol requested all models to perform automatic irrigation scheme triggering irrigation with little moisture stress and unlimited amount of irrigation water supply (Elliott et al., 2015). Although different models vary in their representation (Muller et al., 2019), this setting, which is basically full irrigation, leads to more irrigation application than the reality in the US and most other places in the world. This highlights the need to further improve model representation of irrigation and modelling protocol to apply irrigation in the future. We have accordingly included this information in the revised Methods section.

[Reviewer 3 Comment 9] *Line 118: coefficient of correlation r is not enough to judge a good or bad result. You can have good prediction with low r so I would suggest to add RMSE to your analysis too.*

[Response] We agree with the reviewer that RMSE is a good indicator of variance, but RMSE

also has an issue that it mixed different form of errors (e.g. mean bias, spatial variability) together, making it less efficient to reflect what aspect has been improved or not. We followed this suggestion by taking the Mean Squared Deviation and its Components developed by Kobayashi et al. (2000). The RMSD was decomposed into squared bias (SB), which reflects the different magnitude between the model/reanalysis dataset and the observations, and mean squared variations (MSV), which reflects the consistency in their spatial variations. Accordingly, we found that the SB of CA, GGCMs and the reanalysis were 0.2%, 3.8% and 1.2% for maize and 0.2%, 4.1% and 1.7% for wheat, respectively. So the reanalysis reduced the GGCM simulated biases by ~70%. The MSV of CA, GGCMs and the reanalysis were 2.8%, 1.9% and 1.8% for maize and 5.0%, 2.3% and 2.8% for wheat, respectively. So the reanalysis successfully maintained the good performance of GGCMs in representing the spatial variations of ΔY . These quantitative estimates further help support our conclusions. Thanks for your suggestion!

[Reviewer 3 Comment 10] *Line 144: There is also a temporal variation which could be as much important as the spatial one*

[Response] We agree with the reviewer that the temporal variations are interesting aspects to be explored for irrigation impacts on yield. However, due to data limitations that climate analogue only provides a static pattern, we focus this study on spatial variations. The author team is seeking to explore the temporal variations in a follow-up study. Thanks for the understanding.

[Reviewer 3 Comment 11] *Line 145: precipitation and PET used in those correlations are based on 12 month values or on the months of the crop growing season? If are based on 12 month, I would suggest to use precipitation and PET per growing season instead. I suggest to use more complex climatic indicators that take into account both precipitation and ET for this kind of correlation.*

[Response] Following this suggestion, we used the actual evapotranspiration (AET) and precipitation to perform the same correlation analyses. As you can see from the below figures, although different models show some variations, the general pattern is quite consistent that

the precipitation plays the stronger role than AET in driving spatial variations in ΔY . Thus, our results are robust to the different indicators used. Since actual ET is model-specific and more uncertain than PET, we used PET in the revised manuscript.

Figure R1. Spatial pattern of the correlation between ΔY and precipitation for wheat estimated by different GCMs.

Figure R2. Spatial pattern of the correlation between ΔY and actual evapotranspiration (AET) for wheat estimated by different GGCMs.

Figure R3. Spatial pattern of the correlation between ΔY and precipitation for maize estimated by different GCMs.

Figure R4. Spatial pattern of the correlation between ΔY and evapotranspiration (AET) for maize estimated by different GCMs.

[Reviewer 3 Comment 12] *Line 150-152: what you mean by PET and rainfall variability? Is this spatial variability within each agro-climatic zone or temporal variability per grid? if it is temporal what time period was used?*

[Response] It is the spatial variability, which we have accordingly clarified in the revised manuscript.

[Reviewer 3 Comment 13] *Line 156: again same as before “expanding” can’t reduce yield gap*

[Response] It is corrected accordingly.

[Reviewer 3 Comment 14] *Line 157: The volumetric water demand to reduce the gap is much more interesting than yield gap correlation with PET and rainfall. I would suggest to add volumetric water demand maps or table if possible.*

[Response] We accordingly added the volumetric water demand into supplement Table S1 for major river basins studied.

[Reviewer 3 Comment 15] *Line 158: Using the global river discharge to assess water availability to reduce yield gap is not much relevant for two crops such as wheat and maize. Surface flow might not be available but what about ground water? Also, if water is made available I strongly doubt that it will go to wheat and Maize rather to more cash crops that can justify the extra costs of irrigation investment. Also assuming 20 to 40% of river discharge is used for irrigation is too crude. Instead I would suggest to link water demand to other water risk indicators such as the one produced by the aqueduct project and include baseline water stress, interannual variability, seasonal variability, flood occurrence, and drought severity.*

[Response] Thanks for pointing out the complexities in the issue. We agree with the reviewers that the available water for irrigation may not necessarily go to wheat and maize, due to economic considerations. However, the available surface water resources provide the upper bound of what could potentially be put into wheat and maize production in a sustainable manner that does not mine the fossil water. Therefore, our estimates provide the

conservative estimate on how irrigation could close the yield gap of wheat and maize. At multi-year time-scale, seasonal variability, flood occurrence and drought severity may not necessarily serve as a good indicator, because building water reservoir (e.g. dams) may buffer this short time-scale variability and provide irrigation water, which is a promoted measure in some developing countries. Accordingly, these points have been clarified in the last paragraph of the revised manuscript.

We agree with the reviewer that 20%-40% is a simple assumption for irrigation water availability, but this range of assumptions covered the range of parameters used in the literature. Although this parameter show a large range, choices made do not qualitatively change our results. Therefore, it helps prove that our results are robust to uncertainties associated with the variability in availability of river discharge. We have also considered the aqueduct indices (e.g. baseline water stress and baseline water depletion), which provide additional thoughts on measuring water stress. In particular, the baseline water depletion, which quantifies the ratio of total water consumption to available renewable water supply, is a potential idea to be used for our study. However, we focus on the balance between available river discharge and irrigation needs to close the yield gap, when detailed accounting of other terms of discharge withdraw and consumption is beyond the scope of this study. In the methodological aspect, the ratio-based approach is sensitive to the divided (available river discharge for irrigation). Since we constrain this uncertain term with a large reasonable range, the ratio would thus show a vary large range that would eventually need log transformation, which make it mathematically equivalent to take the balance approach. After careful discussion among the coauthors, we prefer to calculating the balance instead of calculating the ratio. If the reviewer prefer to using the ratio approach, we can fulfill this request. Thanks for the understanding!

[Reviewer 3 Comment 16] *Line 216: remove “to”*

[Response] It is corrected accordingly.

[Reviewer 3 Comment 17] *Line 224: explain better what increments of GDD means. Also can you add a map of the climatic zones of Mueller et al. used in this work?*

[Response] Since this comment is closely related to Reviewer 3 Comment 20, we address them together under the response to Reviewer 3 Comment 20.

[Reviewer 3 Comment 18] *Line 227: Political unit or climatic zone?*

[Response] It is the climatic zone, which we have now described in details following the reviewer's suggestions.

[Reviewer 3 Comment 19] *Line 228: Irrigated map used corresponds to which year? Rainfed wheat is largely used in rotation with other irrigated crops (i.e vegetables) in many parts of the world.*

[Response] The irrigated map used was MIRCA2000, corresponds to around the year 2000, which is around the central of the simulated period. We also agree with the reviewer that crop rotation is one of the limitations of contemporary global crop model simulations. Although many GGCMS can simulate crop rotation, a consistent global map of rotation practices remains not available. We have included this point in the revised Methods section.

[Reviewer 3 Comment 20] *Line 230: what you mean by 100 to 400? Have you used two climatic zones maps? One for GDD and one for precipitation? Please explain more, also the word increments too.*

[Response] We follow this suggestion and the previous comment (Reviewer 3 Comment 17) by clarifying the zoning.

Increments of GDD and precipitation are defined using percentiles of the data. For example, when using 100 climate zones, we use 10 increments of GDD and 10 increments of precipitation. All of the “coldest” observations therefore have GDD < the 10th percentile of GDD across harvested area. The next bin has GDD between the 10th percentile and the 20th percentile, and so on.

The number of zones is somewhat arbitrary, balancing the need to avoid grouping disparate

climate conditions in the same zone and the need to sample wide enough management variation to get a reasonable estimation of attainable yields. We therefore explore sensitivity to the number of zones by examining attainable yields calculated when using GDD x precipitation zones ranging from 10x10 zones (100 total zones), 11x11, 12x 12, ... up to 20x20 zones (400 total zones). These calculations generate 11 values of attainable yields for every grid cell, corresponding with attainable yields estimated for that grid cell using each zonation (10x10, 11x11, etc.). For the primary analysis in this paper, we use the mean of these estimated attainable yields.

We have added maps of the 10x10 and 20x20 climate zones (bounding the range we utilize) for both crops below and in the paper as Supplementary Figure 6. We also clarify that these are equal-area increments of GDD and precipitation to line 244.

[Reviewer 3 Comment 21] Line 243: *Explain in two sentences what parameters were used to calibrate those models and where those data were taken from?*

[Response] The global gridded crop models used here are not specifically calibrated in this study in order to represent the range of uncertainties in model structure and parameters. Many of them do not share the same parameters in the same processes. The uncertainties of the

structures and parameters were represented by this largest-ever global gridded crop model ensemble. We have clarified this with a citation to Muller et al. (2017): “A major difference to field-scale model applications is that at large regional- to global-scale detailed model calibration to field observations is not possible.”

[Reviewer 3 Comment 22] *Line 247: What is harmonon and full harm simulation settings are very specific. I would suggest to limit the description to the unlimited nutrient supply*

[Response] We follow this suggestion by removing the jargon of “full harm”, and focusing on the description to the simulations we used with unlimited nutrient supply.

[Reviewer 3 Comment 23] *Line 258: Instead of using the average across 1980-2010, would it be possible to limit your analysis to a single year i.e. 2010? Doing so, you would be consistent with the data produced by climatic zones, survey data and USDA dataset.*

[Response] We appreciate this suggestion, together with the previous suggestions on exploring the temporal dynamics. However, because ground data-based “climate analogue” do not have enough data to produce the year-to-year variations, we have to work with a static spatial pattern. The average period used here is the common period of data used in “climate analogue” and the temporal period covered by global gridded crop models. We have clarified this choice here in the revised manuscript. Thanks for your understanding.

[Reviewer 3 Comment 24] *Line 271-272: Rephrase*

[Response] We accordingly revise the sentence for clarity.

[Reviewer 3 Comment 25] *Line 315: replace “potential irrigation water withdraw” with “crop water demand” or “irrigation water requirement”*

[Response] We accordingly used “crop water demand” and revise the whole paragraph.

[Reviewer 3 Comment 26] *Line 322: what is the crop specific irrigation efficiencies? The actual irrigation withdrawals is the one measured using flow meters which is not the case here.*

[Response] We agree with the reviewer that using lysimeter measurements can obtain accurate measurements of irrigation efficiencies. However, it is impossible at the moment to upscale lysimeter measurements to the global scale. We therefore used the crop-specific irrigation efficiency produced by Jägermeyr et al. (2015), which is based on LPJmL simulation constrained by AQUASTAT data on irrigation system distribution. To our knowledge, this is the only available global gridded dataset of irrigation efficiency. We have further explained the use of this dataset following this comment.

Reviewers Comments, second round -

Reviewer #2 (Remarks to the Author):

The authors have done a nice job of addressing concerns in my previous review.

Reviewer #4 (Remarks to the Author):

To better estimate the irrigation contribution to wheat and maize yields at the global scale, the manuscript developed a Bayesian framework by integrating the climate analogue and GGCMs, calculated the yield differences between the irrigated and the rainfed wheat and maize (ΔY), and also tested the differences using an independent dataset.

The manuscript has quantitatively improved the understanding of irrigation contribution on wheat and maize yields (e.g., helping yield-gap closure) and analyzed the way (water allocation from river) to increase crop irrigation. I have several additional concerns:

1. The manuscript does not provide any discussion on food security (suggestion: overlapping global hunger/malnutrition information on the result map (ΔY) would make the work more meaningful), nor climate change (extreme events, scenario discussion).
2. Lines 123-125: the authors have compared the results between BMA and CA, why not compared the results between BMA and GGCMs.
3. Lines 134-135: does "the precipitation during the growing seasons" in Supplementary Fig 1 account for cropping issue, say, most agricultural practices in the Yangtze river basin are double cropping.
4. Lines 144-145: please provide possible reason(s) for this inconsistency.
5. Lines 155-157: Since there are 49% for wheat, 48% for maize of their spatial variations of ΔY correlated with PET, it is a bit risky to use the word "mostly".
6. Lines 161-195: In East Asia and South Asia, not like cotton, soybeans and other rainfed crops, paddy rice is usually planted along river. Allocating water from river to wheat and maize irrigation may affect rice planting (important staple food), which is not feasible in many parts of Asia. If allocating water is allowable, it would be costly and needs more coordination, please discuss this.
7. Line 216-217: Since there are relatively large uncertainties somewhere across the globe in Figure 2, informing policymaker directly is risky.

Other minor concerns:

1. Line 63-64: agricultural intensification may not always minimize environmental impacts.
2. Since "climate analogue" and its abbreviation "CA" have been provided in lines 103-104, it is not necessary to use "climate analogue" in the rest of main text (lines 231, 271, 327, 518, 537).
3. Line 117: should be Fig. 1a, d, the same for line 118: Fig. 1b, e; line 123: Fig 1c, f; line 125: Fig. 2a, b
4. It would be useful to have some nations or places' names on the map in Fig.2 or a supplementary map. Although some big nations (the United States, China) are easy to locate on the map, Ukraine (line 133), the Yangtze river basin (line 134) are hard.
5. Line 172: Please provide 30-47% separately for wheat and maize.
6. Line 176: should be southeastern Australia.

Reviewer #5 (Remarks to the Author):

This manuscript presented an interesting result on the maize and wheat yield differences between rainfed and irrigated cropping systems. The authors combined process-based modeling results (GGCMI) using Bayesian model average (BMA) with benchmark of results from a statistical analysis method (CA). The authors analyzed the spatial patterns of yield differences at the global scale and further diagnosed the drivers of those spatial patterns using partial correlation. They

found that precipitation is more like a controller than PET or atmospheric demand of the spatial patterns of yield differences. Finally, the authors analyzed the gap between irrigation water demand and surface water supply and found that found that "30–47% of the contemporary area under rainfed agriculture of wheat and maize cannot achieve the yield gap closure with irrigation from water available in current river discharge, unless more water diversion projects are set in place".

Overall, the topic is interesting to the broader communities. The manuscript is well prepared and organized. As the authors have discussed in their manuscript, there are many limitations and uncertainties in this kind of global analysis and thus we should be more careful when interpreting the numbers or results. I have the following concerns for the authors to consider.

Firstly, I applaud the authors going to the analysis of irrigation water gap in the final part. However, this also opens a big hole for this manuscript, as irrigation is a complex adaptation that contains biophysical and socioeconomic factors. In the modeling world, we definitely can assume irrigation for everywhere to close the yield gap. However, that will never happen in the reality considering the balance of economic cost and gain when adopting irrigation. Therefore, I would not agree that "irrigation demand" is lower as stated in this manuscript.

Secondly, in the CA, process-based models, and gridded US dataset, the yield trends should be very different. For example, there will be no trend about technology improvement in the process-based models. How did the authors deal with this? Is it a fair comparison as shown in Fig. 1?

Thirdly, even with reanalysis as shown in Fig.1, there is a severe overestimation of the yield differences over the US. I guess the overestimation problem should be even worse over other regions of the world due to the differences of management practices and cultivars over rainfed and irrigated cropping areas. How will this overestimation affect the later irrigation water gap analysis?

Finally, the uncertainty information from GGCM and BMA is not fully considered in the final results. I guess the error information given in L51 is from the spatial standard deviations, not really about the BMA uncertainties. The authors should be more serious about the uncertainty quantification given the importance of this topic as well as the numbers they are giving in the manuscript.

Other comments:

It would be helpful to add maps of CA in Figure 2.

L149: is the "deltaY" here from BMA or CA? Could the authors also show the results like Fig. 3 for the CA dataset?

L261-L263: grammar issue with this sentence? You may consider deleting "is" before "that".

L338: wrong citation? That paper is not about PET calculation.

Reviewer #2 (Remarks to the Author):

[Comment] *The authors have done a nice job of addressing concerns in my previous review.*

[Response] We thank the reviewer for the constructive suggestions helping improve the manuscript from its previous version.

Reviewer #4 (Remarks to the Author):

[Comment 1] *To better estimate the irrigation contribution to wheat and maize yields at the global scale, the manuscript developed a Bayesian framework by integrating the climate analogue and GGCMs, calculated the yield differences between the irrigated and the rainfed wheat and maize (delta Y), and also tested the differences using an independent dataset.*

The manuscript has quantitatively improved the understanding of irrigation contribution on wheat and maize yields (e.g., helping yield-gap closure) and analyzed the way (water allocation from river) to increase crop irrigation. I have several additional concerns:

[Response] We thank the reviewer for constructive evaluation and accordingly revise the manuscript. Please find the point-by-point response below.

[Comment 2] *1. The manuscript does not provide any discussion on food security (suggestion: overlapping global hunger/malnutrition information on the result map (delta Y) would make the work more meaningful), nor climate change (extreme events, scenario discussion).*

[Response] Following the reviewer's suggestion, we have provided the malnutrition map (Figure R1) in the supplementary information and added discussions with the spatial information of ΔY . We thank the reviewer for this very constructive suggestion, which help enrich the content of our study. We have also included the names of the major producers for wheat and maize in this map for readers' convenience in locating them in global maps. It should be noted that food security is a much more complex topic than crop production, even though crop production certainly contributes to food security. We have accordingly added a short discussion for the importance of irrigation expansion for the food security, particularly in countries with prevalence of undernourishment. In addition, we have added a short discussion that future changes in climate will not only affect ΔY but also water availability and will thus have to be considered in future assessments with the references of Rosa et al. (2020). However, since this analysis is based on the historical period, we refrain from discussing specific climate scenarios.

Thanks for the constructive suggestions.

Figure R1. Spatial distribution of prevalence of undernourishment (%) during 2000-2010 according to Food and Agriculture Organization of the United Nations (data source: <https://unstats.un.org/sdgs/indicators/database/?indicator=2.1.1>). Major producers are delineated in black and named for readers' convenience.

[Comment 3] Lines 123-125: *the authors have compared the results between BMA and CA, why not compared the results between BMA and GGCMs.*

[Response] We have accordingly added here comparison between BMA results and original model GGCMs as suggested.

[Comment 4] Lines 134-135: *does “the precipitation during the growing seasons” in Supplementary Fig 1 account for cropping issue, say, most agricultural practices in the Yangtze river basin are double cropping.*

[Response] The growing season we used in this study came from MIRCA dataset, which has considered the double cropping, if the major practice in a region is double cropping. This is exactly the case for the lower reach of the Yangtze River basin, where wheat-rice double cropping is prevalent.

[Comment 5] *Lines 144-145: please provide possible reason(s) for this inconsistency.*

[Response] Thanks for the suggestion. The most possible reason is that the yield over this region is more constraint by fertilizer managements than by irrigation (Mueller et al., 2012). We have accordingly added this information here following the suggestion.

[Comment 6] *Lines 155-157: Since there are 49% for wheat, 48% for maize of their spatial variations of delta Y correlated with PET, it is a bit risky to use the word “mostly”.*

[Response] We have accordingly revised the wording here from “mostly” to “more”, which better reflects the relative role of different climatic factors.

[Comment 7] *Lines 161-195: In East Asia and South Asia, not like cotton, soybeans and other rainfed crops, paddy rice is usually planted along river. Allocating water from river to wheat and maize irrigation may affect rice planting (important stable food), which is not feasible in many parts of Asia. If allocating water is allowable, it would be costly and needs more coordination, please discuss this.*

[Response] Thanks for the constructive suggestion. We follow this suggestion by including a discussion on the difference between irrigation for wheat and maize and irrigation for rice over Asia, which has additional functional features such as facilitating transplanting and drainage for facilitating tillering (Line 307-311). Our analyses, however, was not affected since we only allocate contemporary river discharge that was not used for any irrigation or human purposes.

[Comment 8] *Line 216-217: Since there are relatively large uncertainties somewhere across the globe in Figure 2, informing policymaker directly is risky.*

[Response] We agree with the reviewers that the uncertainties are indeed critical for policymakers to consider when using any of the data streams (CA, GGCMs and the BMA reanalysis). We have highlighted that policy making should consider the uncertainties in the data as suggested by the reviewer.

Other minor concerns:

[Comment 9] 1. Line 63-64: *agricultural intensification may not always minimize environmental impacts.*

[Response] We have revised the sentence to indicate the needs for sustainable intensification pathways.

[Comment 10] 2. *Since “climate analogue” and its abbreviation “CA” have been provided in lines 103-104, it is not necessary to use “climate analogue” in the rest of main text (lines 231, 271, 327, 518, 537).*

[Response] Corrected accordingly.

[Comment 11] 3. *Line 117: should be Fig. 1a, d, the same for line 118: Fig. 1b, e; line 123: Fig 1c, f; line 125: Fig. 2a, b*

[Response] Corrected accordingly. Thanks!

[Comment 12] 4. *It would be useful to have some nations or places’ names on the map in Fig.2 or a supplementary map. Although some big nations (the United States, China) are easy to locate on the map, Ukraine (line 133), the Yangtze river basin (line 134) are hard.*

[Response] In the revised supplementary Figure 5, we have accordingly delineated the names of major wheat and maize producers on the map in order to facilitate readers to identify them.

[Comment 13] 5. *Line 172: Please provide 30-47% separately for wheat and maize.*

[Response] We have clarified that this range is for different thresholds considered, not for different crops.

[Comment 14] 6. *Line 176: should be southeastern Australia.*

[Response] Corrected accordingly, thanks.

Reviewer #5 (Remarks to the Author):

[Comment 1] *This manuscript presented an interesting result on the maize and wheat yield differences between rainfed and irrigated cropping systems. The authors combined process-based modeling results (GGCMI) using Bayesian model average (BMA) with benchmark of results from a statistical analysis method (CA). The authors analyzed the spatial patterns of yield differences at the global scale and further diagnosed the drivers of those spatial patterns using partial correlation. They found that precipitation is more like a controller than PET or atmospheric demand of the spatial patterns of yield differences. Finally, the authors analyzed the gap between irrigation water demand and surface water supply and found that found that “30–47% of the contemporary area under rainfed agriculture of wheat and maize cannot achieve the yield gap closure with irrigation from water available in current river discharge, unless more water diversion projects are set in place”.*

Overall, the topic is interesting to the broader communities. The manuscript is well prepared and organized. As the authors have discussed in their manuscript, there are many limitations and uncertainties in this kind of global analysis and thus we should be more careful when interpreting the numbers or results. I have the following concerns for the authors to consider.

[Response] We thank the reviewer for the positive evaluation and constructive suggestions. We have accordingly revised the manuscript following the suggestions. Please see the point-by-point response below.

[Comment 2] *Firstly, I applaud the authors going to the analysis of irrigation water gap in the final part. However, this also opens a big hole for this manuscript, as irrigation is a complex adaptation that contains biophysical and socioeconomic factors. In the modeling world, we definitely can assume irrigation for everywhere to close the yield gap. However, that will never happen in the reality considering the balance of economic cost and gain when adopting irrigation. Therefore, I would not agree that “irrigation demand” is lower as stated in this manuscript.*

[Response] We agree with the reviewer that the application of irrigation in reality was under constrain by economic costs and gains. We have accordingly clarified in the revised manuscript that the irrigation demands we considered were the potential irrigation demands to close the yield gap without considering economic constraints. The economic constraints, as well as competition with growing water needs for industrial and domestic sectors are beyond the scope of this study. But we follow the reviewer suggestion to improve the discussions on this topic and the implication of our study for the modelling in the hydrology-agriculture-economic nexus (e.g. Rosa et al., 2020), in which irrigation needs are critical factors. Thanks for the suggestion!

[Comment 3] *Secondly, in the CA, process-based models, and gridded US dataset, the yield trends should be very different. For example, there will be no trend about technology improvement in the process-based models. How did the authors deal with this? Is it a fair comparison as shown in Fig. 1?*

[Response] The simulation by process-based models, gridded US dataset and CA approach have similar temporal coverage, which has been detailed in the Methods section. We agree with the reviewer that yield trends could potentially affect the comparison between different approaches for interannual variability and decadal changes, therefore, we used the common periods covered by the datasets to minimize the effects and analyze only the average ΔY over the study period in the manuscript. This static analysis thus should not be largely affected by yield trends. Thanks for your careful examination and understanding.

[Comment 4] *Thirdly, even with reanalysis as shown in Fig.1, there is a severe overestimation of the yield differences over the US. I guess the overestimation problem should be even worse over other regions of the world due to the differences of management practices and cultivars over rainfed and irrigated cropping areas. How will this overestimation affect the later irrigation water gap analysis?*

[Response] We agree with the reviewer that the reanalysis ΔY is still larger than the USDA statistics because the GGCM protocol requested all models to perform automatic

irrigation scheme triggering irrigation with little moisture stress and unlimited amount of irrigation water supply (Elliott et al., 2015). Although different models vary in their representation (Muller et al., 2019), this setting leads to more irrigation application than in reality in the US. In a follow-up study with GGCMs (Yu & Wang et al., *in prep*), when we compared with the simulations with irrigation experiments that applying similar fully irrigation conditions, we found that models are not biased for maize (mostly concentrated in US) though models tend to overestimate ΔY for at least some wheat trials (Figure R2), further supporting that the differences with USDA were mainly driven by the deficit irrigation practice. Since the yield gap analysis is to achieve attainable yield under well irrigated condition, this issue would not create biases. To follow the reviewer's suggestion, we have accordingly included this information in the revised Methods section, highlighting the need to further improve model representation of irrigation and modelling protocol to apply irrigation in the future in order to suit more diversified types of irrigation applications.

Figure R2 Comparison between simulation by GGCMs and irrigation experiments. The lower panel shows spatial distribution of irrigation experiments collected from Web of Sciences (red sites for wheat, black sites for maize and blue sites for rice).

[Comment 5] Finally, the uncertainty information from GGCM and BMA is not fully considered in the final results. I guess the error information given in L51 is from the spatial standard deviations, not really about the BMA uncertainties. The authors should be more serious about the uncertainty quantification given the importance of this topic as well as the numbers they are giving in the manuscript.

[Response] We thank the reviewer for the suggestion, which was also the concern by one of the previous reviewers. The uncertainty reported throughout the manuscript had been changed to reflect the model ensemble/BMA reanalysis. We agree with the reviewer that uncertainties are still large, but this provides a good assessment on global ΔY uncertainties, which has not been studied before. To better follow the reviewer suggestion,

we have carefully discussed the source of this uncertainties in the later parts of the manuscript.

Other comments:

[Comment 6] *It would be helpful to add maps of CA in Figure 2.*

[Response] Following the reviewer’s suggestion, we provided the corresponding map of CA in Figure R3. However, since the evaluation with USDA statistics (Figure 1) shows that CA is less robust than GGCMs/BMA in the spatial variations, spatial variations of ΔY should be better studied with BMA/GGCMs. we believe this is not the best spatial information we should show to the readers, thus we put it as a supplementary Figure.

Figure R3. Spatial distribution of ΔY from climate analogue approach for (a) wheat and (b) maize over contemporary growing area.

[Comment 7] *L149: is the “deltaY” here from BMA or CA? Could the authors also show the results like Fig. 3 for the CA dataset?*

[Response] We have accordingly clarified that it is from reanalyzed ΔY . Following the reviewer suggestion, we mapped the same figure here with ΔY from CA. As Figure R4 shows, our conclusion remains unchanged with this analysis. As the spatial variations in ΔY from CA were interpolated based on simple climatic indexes which do not represent the best spatial information we should show to the readers, and it provides no additional information to current figures, we do not include it in the manuscript. Thanks for your understanding.

Figure R4. Partial correlation in the spatial domain between ΔY from the climate analogue approach and climatic variables (potential evapotranspiration (PET) and mean annual precipitation (MAP)) for wheat (top panel) and for maize (bottom panel).

[Comment 8] L261-L263: *grammar issue with this sentence? You may consider deleting “is” before “that”.*

[Response] Corrected accordingly. Thanks for the careful examination.

[Comment 9] *L338: wrong citation? That paper is not about PET calculation.*

[Response] We have accordingly replaced the correct citation of Haude equation. Thanks.